# Multi-Purpose Accessibility of Mountain Area Forests for the Purpose of Forest Management and Protection of the State Border

**Doroteja Krivić-Tomić [1], Ivica Papa [1,*] and Mijo Kožić [2]**

1   Department of Forest Engineering, Faculty of Forestry and Wood Technology, University of Zagreb, Svetošimunska Cesta 23, 10000 Zagreb, Croatia; krivic.dorotea@gmail.com
2   Croatian War College "Ban Josip Jelačić", Ilica 256b, 10000 Zagreb, Croatia; mijo.kozic@morh.hr
*   Correspondence: ipapa@unizg.sumfak.hr

**Abstract:** The planning and implementation of surveillance of state territory in forested border areas, especially mountainous areas, is considered to be highly complex. This is illustrated by the example of the difficulties the European Union faced in controlling the 2015 European migration crisis. Thereby, Croatia has the difficult task of protecting the borders of the Union because a particular problem on the Western Balkan Route is the so-called bottleneck to Slovenia in the area of the Municipality of Donji Lapac, which consists of the green border with Bosnia and Herzegovina. Consequently, by using the example of planning multi-purpose forest roads, the aim of this paper is to propose the inclusion of the road network of border management units of mountain areas in the control system of the state's green border, which, in this paper, includes its surveillance and protection by land for the purpose of national security. The research was conducted on the example of the Visočica–Lisac border management unit in the Municipality of Donji Lapac. The results of the research indicate a possible solution to the control of the border management unit by establishing a two-level surveillance system. The higher level consists of strategically defined surveillance points and corresponding multi-purpose forest roads designed on a tactical level. At this level, the priority is protection or, more precisely, defense of the state border. The lower level consists of tactically determined surveillance points with corresponding multi-purpose forest roads designed on the operational level. In addition to protecting the state border, this level would also have the task of protecting the forest, that is, monitoring the area of the management unit.

**Keywords:** forest roads; Western Balkan Route; European migration crisis

## 1. Introduction

Migration, especially illegal migration, is one of the biggest problems of the modern world [1] because the contemporary understanding of borders is bipolar. Namely, "[...] the semantics of borders reveals a dialectical relationship between a completely 'fixed borderline' [...] and a 'vague border area' [...]" [2] (p. 294). In other words, ref. [3] states that fixed, impassable barriers were built for national security purposes. Therefore, there is no doubt that they are considered a contemporary iron curtain [4] (p. 112). However, an issue is that borders are conventional, mentally imagined lines. They consist of intentionally placed border crossings and imagined bounded areas, the so-called *buffer* zones around them, which together form the porous border area [5] (p. 5). In the context of the 2015 European migration crisis, an example of such a border is the concept of the green-grey border [6], which exists between Bosnia and Herzegovina and Croatia.

Namely, the Balkans have been an area of migration between Europe, Asia, and Africa for centuries [2] (p. 294), whereby the final destination for most migrants is the countries of Western Europe [1] (p. 33). Therefore, disregarding the actual geographical and cultural borders of the Balkan Peninsula, the countries encompassed by this name are considered to

be the border area of Western Europe [7], so it is not surprising that the Western Balkan Route is one of the eight main migration routes to the European Union [8,9]. Consequently, the Union has been facing the phenomenon of so-called *transit migration* [10] in recent years.

The Western Balkan Route consists of a journey by sea to Turkey or Greece, by land to Macedonia or Bulgaria, and further to the European Union, through Serbia and Hungary, Serbia and Croatia, or Bosnia and Herzegovina and Croatia. The route was a secret at first. It was used to smuggle people through mountain terrain and forested areas of Southeast Europe to the desired country in the West [11]. The route is/was also preferred for drug and arms trafficking [1]. In 2015, after a large migrant wave towards the Austrian-Hungarian border, the route became a controlled transit corridor from Greece to Austria. After Hungary and Bulgaria completed the construction of wire fences on the focal parts of their borders in 2016, the corridor was closed. The migrant wave has developed a new strategy for entering the territory of the Union. The route was modified across Croatia's green border with Bosnia and Herzegovina [8] (p. 193). Since it is one of the longest-established land borders in Europe, Croatia, as a member of the Union, has been given the difficult task of safeguarding European sovereignty and cultural identity. Therefore, in addition to the national security system, this Union border is also under strong monitoring by the European Border and Coast Guard Agency (Frontex). As this is a predominantly green border between the two countries, this paper will provide an example of how the forestry profession can contribute to the improvement of Croatia's homeland security system. In other words, the aim of the paper is to show how the planning of multi-purpose forest roads in the road network of border management units in mountain areas can support the established system of monitoring the state's green border. Consequently, the paper starts with (a) the idea of determining the key points from which it is possible to monitor the management unit, (b) the idea of determining monitored cardinal points within the network of forest roads in the management unit, and (c) the idea of designing multi-purpose forest roads that, on the one hand, would serve for the needs of forest management and, on the other hand, for the protection (that is, surveillance and defense) of the state border.

## 2. Overview of Previous Research

In general, the security risk in forested areas can be broken down into traditional and extended understandings. According to the former, in addition to the risk of fire, it also refers to poaching, theft of wood assortments from landing sites, and illegal tree felling [12] (p. 158). A broader understanding of the security risk in forest areas is mainly related to the issue of national security. In this context, it can be said that it refers to the defense of the state border, illegal crossing of the green border, and various types of smuggling through the forest. Safety risks also include destructive activities of *off-road* motorcycling and illegal races with *quad* vehicles [13] (p. 1). Therefore, when it comes to the land control of forested (especially border) areas, the primary objective is to monitor forest transport infrastructure.

During the wartime period in the first half of the 20th century, forest areas were monitored by establishing a road maze and controlling road junctions and roads within forest complexes [14–16]. The movement of vehicles of any kind outside the existing transport communication could have been excluded with a high probability [14]. Likewise, in his opinion, the existing road network was also preferred when walking, and pedestrians, if they were away from the roads, at some point certainly strived to approach them. However, [14] (p. 31) does not agree that the forest road network should look like a maze, so he suggests organizing circular traffic. He believes that circular roads prevent congestion on roads in the forest and reduce the possibility of unnoticed movement for those who want to remain unnoticed.

Ref. [17], in their paper, emphasize that today's forest roads should strive for multi-functionality. Therefore, the order of importance of parameters, on the basis of which forest road routes are planned, has changed over time. According to them, forest road planning "[in] the past took into account only hydrology and slope factors. Today, socioeconomic factors are also included [. . .]" [17] (p. 18). Amani (2000) is of a similar opinion, arguing that

multi-purpose forest roads condition multi-purpose forest management (as cited in [17] (p. 13)). His intention is to state that, in addition to the prevailing practice of emphasizing the economic component, the political component should also be taken into account when managing forests in certain areas. Such an opinion is shared by Potočnik (1996) and Bjorklund (2006), as they state that national policy is also taken into account in forest management (as cited in [18] (p. 5389)). The significance rate of forest roads for security use in Slovenia is 3.09%, of which 1.79% refers to the police and 1.30% to the army [19] (p. 68).

By examining previous research on this topic, it can be established that the planning of the optimal network of primary forest transport infrastructure for the purpose of control (which includes surveillance and protection) of border areas is increasingly based on multi-criteria decision-making on the route of the future forest road. Most often used are the Delphi method, the Analytic Hierarchical Procedure, or the Fuzzy Set Theory [20] (p. 1768), which are based on weighting or scoring. In addition to multi-criteria assessments, it is important to mention that computer and linear programming [21] (p. 63) are also used in the planning of forest transport infrastructure because, in this manner, a complete overview of the most favorable position of conceptual routes in space is obtained. The collective name for such multi-criteria methods is *spatial multi-criteria evaluation* (SMCE) [20] (p. 1768). When planning the detailed layout of the road network in extremely large areas, ref. [22] (p. 1) explains that "[...] optimization of the layout of the forest road network can also be based on iterative algorithms [...]. This belongs to complex methods because, in addition to linear programming, it is also hierarchical and takes into account not only the issue of road construction but also the issue of timber harvesting.

When it comes to forestry surveillance of forest transport infrastructure, aerial photogrammetric or satellite images are less commonly used because they cannot always show in detail what is occurring on the ground due to the structure of the stand, that is, the degree to which the canopy covers the ground. Therefore, the so-called terrestrial methods are mostly applied, and they most often refer to wireless sensor networks [23] (p. 2). For example, in order to detect the location of illegal tree felling, sound sensors for the detection of chainsaw operation and vibration sensors for the detection of tree fall can be installed in forests [23] (p. 2). Furthermore, for the purpose of preventing theft of wood assortments, a surveillance system based on an accelerometer, geophone, and magnetometer is used, which provides information on the entry point of the vehicle into the forest area and the speed and direction of its movement [13] (p. 1). Similarly, an acoustic system used for the surveillance of the remote transport of wood and the movement of forestry officials' vehicles in the management unit is also applied. This system consists of a database of sounds of company and subcontractor vehicles in the process of timber harvesting [12] (pp. 158–162).

When monitoring the green border in order to prevent illegal crossings, the conventional surveillance system based on security checkpoints and patrols at specific time intervals is still widely used [24]. Therefore, one of the most radical examples of preventing illegal border crossings is still the application of the US Border Security Policy paradigm called *Prevention Through Deterrence* (PTD). In this case, border control uses forest transport infrastructure mapping technology. In other words, "[...] maps are made that distinguish between 'controlled' or enclosed parts and 'monitored' or unenclosed parts" [25] (p. 163). In more detail, in order to ensure the safety of urban zones in border areas, dedicated forest transport infrastructure planning directs migrants towards more remote (rural) areas, where environmental conditions act as a natural barrier to movement. On the one hand, the *spatial displacement effect* [26] is achieved in this manner, while, on the other hand, the implementation of the so-called *Tactical Advantage Act* [25] is enabled.

A step prior to mostly applying contemporary technology to green border surveillance is to combine conventional methods with sensor technology. In this case, surveillance consists of (a) *border patrol*, which controls a certain part of the border day and night; (b) *ambushes*, that is, patrols whose task is to prevent illegal crossings at night; (c) *keeping watch*, which refers to the daily surveillance of the border and terrain of the neighboring

country through a system of observation posts; (d) stationary or mobile *thermal cameras*, which enable the detection of targets, recognition, and observation in all weather conditions; and (e) the application of a stationary or mobile *Askarad* radar system for monitoring, finding and classifying moving targets, and precisely locating them [27] (pp. 1395–1396).

An example of linear architecture in contemporary transport infrastructure surveillance technology for illegal green border crossings is the *Unattended Ground Sensor Network* (UGS). Namely, according to estimates, in a five-year period, 1000 migrants create an average of 722 m of forest trails, i.e., degrade 656 m$^2$ of forest land [28] (p. 403). Therefore, this network is suitable for preventing border crossings in areas remote from local border crossings. "[The] UGS network was developed to remotely detect, localize, identify, and classify targets for the purpose of [...] situational awareness, perimeter protection, border control, and surveillance of borders and targets" [29] (p. 2). To put it simply, the network consists of sensors hidden in the field that channel the data to the nearest central node (base) over long distances, and the information thus collected from all nodes is further sent to the center. The sensors can be magnetic, acoustic, infrared, radar, or capacitive. The UGS network is further connected to the *Intelligent Route Surveillance*—IRS system, based on a triple traffic classification: (a) non-motorized traffic (people and livestock), (b) small motorized traffic (motorcycles, cars, and vans), and (c) large motorized traffic (trucks and forest wheel and track machinery) [29] (p. 6).

However, for the purpose of monitoring forest transport infrastructure in mountainous areas, [30] (pp. 0849–0851) believes that the most suitable method is MWSN (*Multimedia Wireless Sensor Nodes*). According to this method, the area of the green border covered by the sensor node is called the *Field of View*—FOV. Each sensor node consists of (a) a *sensor unit* with a thermal camera for capturing images in space; (b) a *communication unit* that enables communication of the sensor node with other nodes in the network and sending information from the environment to the desired destination (e.g., the nearest police station); (c) a *power unit*, i.e., a battery; and (d) an *information processing and storage unit*. The principle of operation of the system is such that each node has the task of protecting the neighboring (adjacent) sensor node. Namely, in case of damage to the node, the next one, which is on the same route but in a different place, can replace the damaged node [30] (p. 0852).

Nevertheless, the advancement of sensor technology is aimed at creating a hierarchical architecture. One example of such a method of controlling forest road infrastructure is the platform for military surveillance of green borders, *FemtoNode*. The system consists of two sensor levels. The nodes of the so-called low sensors are scattered along the borderline, either on the ground or underground. At the moment when a vehicle crosses the borderline, the low sensors sound an alarm to activate the so-called high sensors, i.e., *Unmanned Aerial Vehicles*—UAVs equipped with, for example, vehicle recognition radars [30] (p. 0849).

An even more complex hierarchical architecture surveillance system is the application of the hybrid architecture of wireless sensor networks. An example of this is the *BorderSense* method with three layers of hierarchy, that is, three types of sensor nodes. The first are *scalar sensor* nodes equipped with a seismic sensor and arranged on the ground or underground. The second type consists of *multimedia sensor nodes* equipped with video cameras or night vision binoculars located on observation posts along the green part of the border. The last are *mobile sensor nodes*, i.e., mobile ground robots and unmanned aerial vehicles [24] (p. 468). This method is characterized by a high level of reliability, i.e., a minimum number of false alarms.

The technologies and methods used in the Croatian homeland security system are mostly classified and not publicly available. Therefore, on the basis of very limited documentation, it is only possible to assume that Croatia is in the stage of combining the application of conventional methods of green border control with the use of a linear architecture of various sensors. Consequently, in order for the primary forest transport infrastructure to be used for multiple purposes, it is desirable that its planning be based not only on forestry multi-criteria assessments of its optimization but also on the possibility of applying more complex methods of security systems for its surveillance.

### 3. Methodology

The area of the Municipality of Donji Lapac is one of the high-frequency crossing zones on the border between Bosnia and Herzegovina and Croatia. Namely, this part of the Croatian territory is the shortest route to Slovenia, which, at the time of the 2015 European migration crisis, was the Schengen border. In order to avoid border control, migrants stopped moving along the gray border, so demanding forested areas often became alternative routes. Therefore, passing through this part of the Western Balkan Route was called a "gamble" [31,32]. Nevertheless, within forest complexes, migrants will still resort to moving through forest transport infrastructure, at least part of the way. Consequently, the aim of this paper, based on the example of planning multi-purpose forest roads, is to provide a proposal for the inclusion of the road network of border management units in mountain areas in the surveillance system of the green border of the state.

The research starts with the following research questions:

1. Can the state border, as well as the territory of the management unit, be controlled if the strategic surveillance point is positioned on the highest peaks of the management unit?
2. Does the surveillance of forest roads within the management unit depend on defining the position of the monitored cardinal points and terrain relief?
3. Does the multi-purpose forest road designed on a tactical level affect the accessibility of the management unit, although this is not its primary purpose?
4. Does the length of the multi-purpose forest road designed on the operational level affect the size of the accessible area in the management unit?

The analysis was conducted on the example of the Visočica–Lisac management unit bordering Bosnia and Herzegovina. Within the framework of forest management, it is under the jurisdiction of the company Hrvatske šume d.o.o., i.e., the Forest Administration of Gospić and its component Forest Office of Donji Lapac. It is positioned between 44°28′40″ north latitude and 15°57′45″ and 16°08′25″ east longitude [33] (p. 11).

The total area of the Visočica–Lisac management unit is 4774.37 ha [33] (p. 11). The management unit is divided into 91 compartments with associated sub-compartments. The relief is characterized by mountain ridges, very steep slopes, karst fields, and deep ravines. In terms of vegetation, this management unit is characterized by uneven-aged beech forests and uneven-aged Austrian oak forests, i.e., the following forest communities: Austrian oak and Southern European flowering ash forest (*Fraxino orni-Quercetum cerridis* Stefanović 1968), beech forest with autumn moor grass (*Seslerio autumnalis-Fagetum sylvaticae* (Horvat) M. Wraber ex Borhidi 1963), mountain beech forest with large red dead nettle (*Lamio orvalae-Fagetum* (Horvat 1938) Borhidi 1963).

The road network has a total length of 58.91 km, of which 47.1 km are forest roads and 11.8 km are public roads. Given that 38.59 km is entered into the calculation of the accessibility of this management unit, its total accessibility is 8.07 km/1000 ha [33] (p. 84). Taking into account the minimum required classical accessibility of mountain areas of 15 m/ha from 1990 or 20 m/ha from 2012, as stated by [34], the aforementioned indicator of total accessibility leads to the conclusion that this management unit is poorly accessible.

The research is based on the M 1:25,000 topographic map and a digital relief model with a resolution of 15 × 15 m of the Visočica–Lisac management unit. First defined on the topographic map are strategic and tactical surveillance points. Strategic surveillance points are those positions from which it is possible to control the state border. They are based on the idea of a location to which the "Lisac" forest road leads, which was built in 2020 as a result of cooperation between the company Hrvatske šume d.o.o. and the Ministry of the Interior for the purpose of state border control. Considering that the peak to which the aforementioned road leads is the highest point of the southern part of the management unit, it is to be assumed that it has strategic significance because a stationary video surveillance system has been installed there. Furthermore, taking into account that, on Gologuz Peak (one of the highest in the northern part of the management unit), there is a transmitting facility, in this paper, it is defined as the second strategic

surveillance point (Figure 1). Tactical surveillance points were defined based on the research of [35]. He uses the *photographic detection system*, hidden at road intersections, to provide the US Forest Service with information on the number and type of vehicles passing through forest complexes. In accordance with the above, the paper starts with the thesis that the tactical (planned and installed by foresters) points in the Visočica–Lisac management unit are defined with regard to the cardinal points in its forest road network. The proposal is that, from these positions, it is possible to control (a) the entry/exit points of the management unit and (b) the key intersections of the main and side forest roads in the transport network. The intersections were selected based on the position of the roads leading to/from the management unit. Consequently, the peaks of Pod Barićev at 869.75 m above sea level, in the area of Visočica (Tactical Point 1) and of Špija at 804.33 m above sea level, in the area of Lisac (Tactical Point 2) were selected as tactical surveillance points. The assumption is that, from them, it is possible to view the largest number of monitored cardinal points (Figure 1).

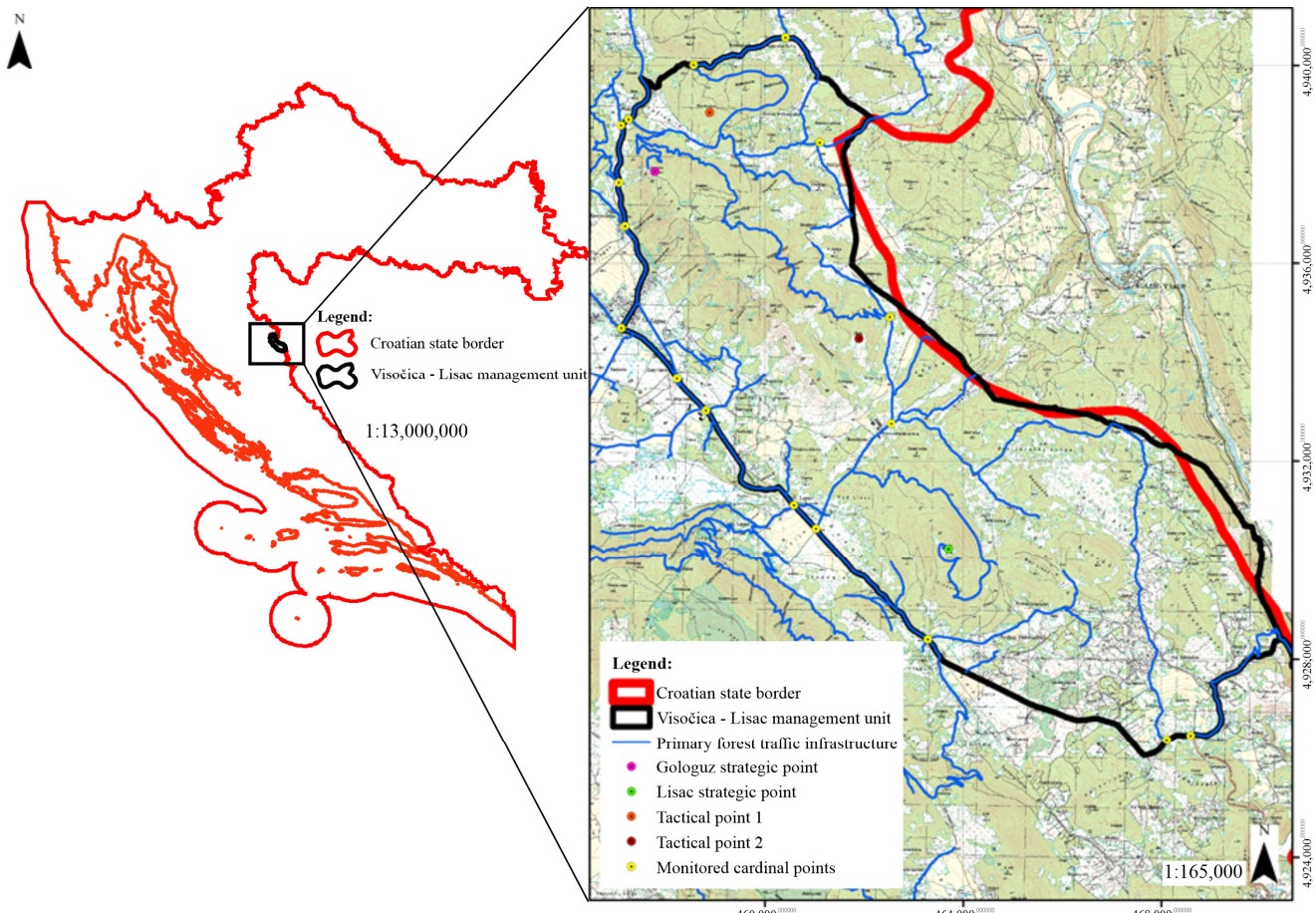

**Figure 1.** Surveillance points in the Visočica–Lisac management unit. Source: Prepared by the author.

The surveillance of the management unit, from the strategic and tactical points defined in this manner, was simulated on the basis of data on the stationary video surveillance system manufactured by FLIR Systems (*the stationary video surveillance system was acquired by Croatia in a public procurement procedure for the defense and security of the state. Therefore, the technical specifications are classified in accordance with the Croatian Data Confidentiality Act (Official Gazette NN 79/07, 86/12). For the purposes of this research, we were allowed to use them, but not to make them publicly available*), which Croatia uses to control its border in the Municipality of Donji Lapac.

When it comes to the analysis of forest transport infrastructure, thereby taking into account that the strategic planning of forest roads, according to [34], implies the level of the forest management area, and the tactical planning, the level of planning in the management unit, forest roads leading to strategic surveillance points are, in this paper, understood as tactically designed routes. On the other hand, conceptual roads that would lead to tactical points of visibility are, in this paper, understood as operationally designed routes. These are individual forest roads that, in addition to surveillance, primarily serve to make accessible a smaller area within the forest complex. Consequently, routes designed on the tactical level of planning are, in this paper, referred to as tactical (multi-purpose) forest roads, while those designed on the operational level are referred to as operational (multi-purpose) forest roads.

Segments of the horizontal and longitudinal development of tactically designed forest roads were analyzed on the basis of *Glavni projekt šumske ceste Lisac* (main/implementing project design of Lisac Forest Road). The planning of conceptual routes for operational forest roads was carried out by designing a zero-line on the Croatian Basic Map (HOK) at a scale of 1:5000 with an equidistance of 5 m between contour lines using the ArcMap 10.1 program (Esri, Redlands, CA, USA). Careful consideration of laying a zero-line polygon of future routes is a major factor in the planning of the forest road network [36] (p. 2456). The starting point of the conceptual route was positioned on the route of the existing primary forest transport infrastructure, while the endpoint was the tactical control point (the elevation at which it was defined). Based on their distance from each other and the altitude difference between them, the average slope of the zero-line polygon was calculated. Then, the so-called divider segment was determined, that is, "[. . .] the value that represents the constant distance between the contour lines for a certain slope" [37] (p. 63). In doing so, zero-line polygons of conceptual routes are conditioned by a maximum longitudinal slope of 12% to (exceptionally, also for shorter distances) 14%, which has been defined for such terrain by [38].

## 4. Research Results

The analysis of visibility from strategic surveillance points showed that the Lisac strategic point in the Visočica–Lisac border management unit is defined on the basis of situational awareness. Despite the fact that the sizes of the set parameters on the actual camera are not known, for the purposes of this research, the simulation of sophisticated video surveillance from this point assumed the setting of the parameters of the thermal imaging camera for detection, recognition, and identification of vehicles at maximum values for selected heights of 4 m (which is the height of a forest truck, trailer, or truck unit) and 8 m (twice the height of a truck, trailer, or truck unit). Consequently, it was concluded that, from this point, it is possible to control the state border and half of the management unit. Whether the video surveillance parameters are defined at a height of 4 m or 8 m, detection and recognition of vehicles from this point are possible in equal quality. As expected, the visibility field for vehicle identification is lower in both examples, as the maximum values of the parameters are also lower (Figure 2). This leads to the conclusion that, in this part of the management unit, Croatia has applied prevention by deterring illegal entry into the country.

However, when it comes to the Gologuz strategic point, the analysis shows negative results for the possibility of state border surveillance. By setting the visibility parameters to maximum height values of 4 m and 8 m, it was observed that it is not possible to control either the state border or the management unit from Gologuz Peak (Figure 3). Furthermore, it is important to point out that, from this point, the surveillance of the area of Visočica itself is limited. This is due to the greater circuity of the relief in this part of the management unit on the one hand and the partial overgrowth of the peak on the other. However, the analysis found that, from this point, the place of Donji Lapac, its access roads, and the hamlets on the west side can be monitored. This leads to the conclusion that Gologuz Peak is suitable for surveillance deep into the territory of Croatia.

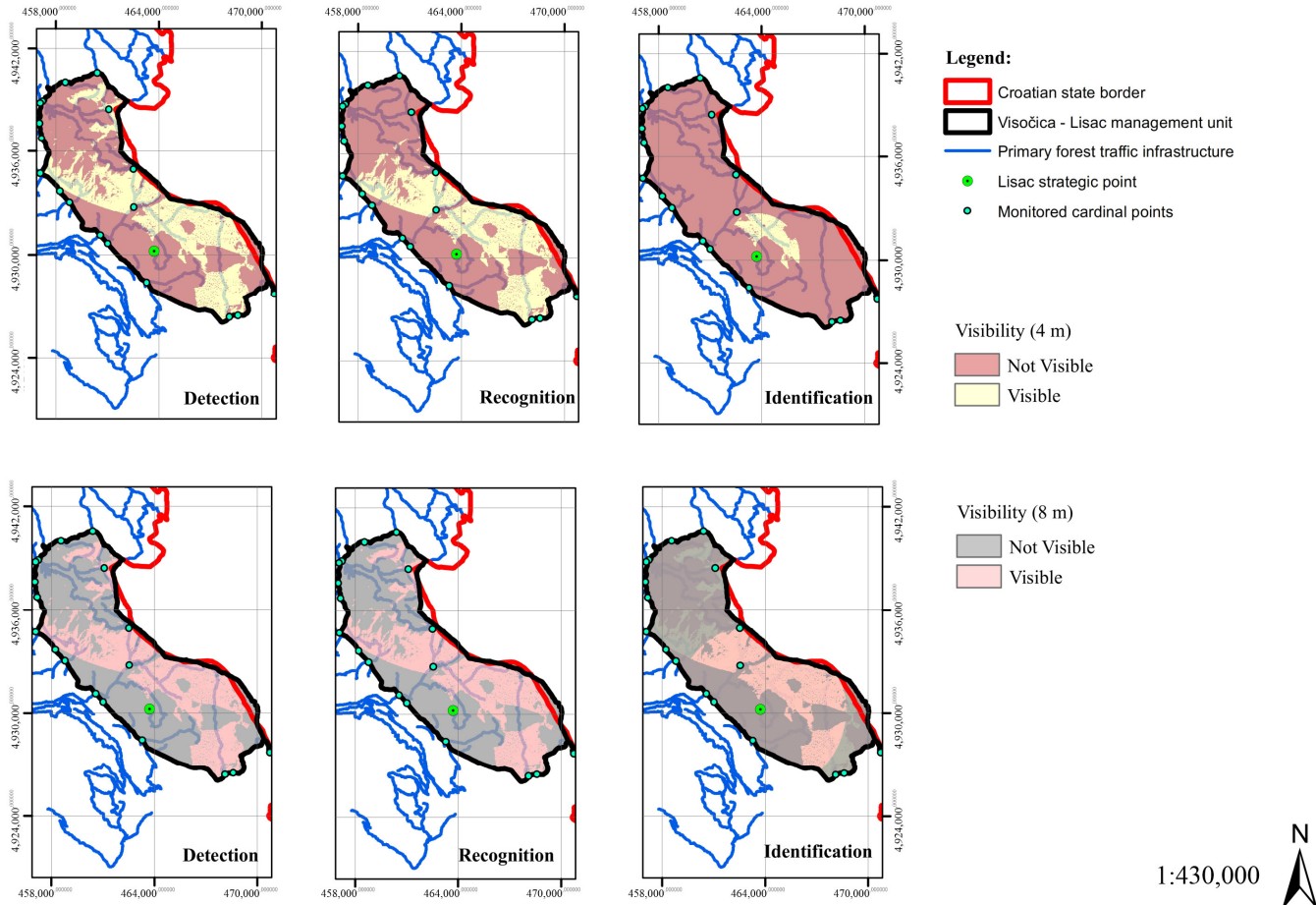

**Figure 2.** Visibility from the Lisac strategic point by simulating the installation of a thermal imaging camera at a height of 4 m and 8 m. Source: Prepared by the authors.

Based on the above, it can be concluded that height does not have as much influence on the size of the monitored area as does the circuity of the terrain. Namely, the area of Lisac is less hilly compared to Visočica. Therefore, the Lisac Peak dominates the area, so the visibility from that point is better. On Visočica, the terrain is more circuitous, so Gologuz Peak is surrounded by a number of other elevations, which, consequently, interfere with the width of the view from that point. In addition to the above, the overgrowth of the peak should also be taken into account. While clear-cutting was carried out on Lisac, Gologuz is partially overgrown.

An analysis of tactical forest roads (Figure 4) showed that the structure of primary forest roads is actually side forest roads. In [39], it is stated that it starts as a road branch from the previously existing Borićevac–Lisac forest road. From 730 m above sea level to the very peak (997 m above sea level), the road overcomes an altitude difference of 267 m with an average longitudinal slope of 8%. The route ends at chainage 33 + 77.25 hm. This road passes through compartments/sub-compartments 57a, 57b, 57d, 59a, 60a, and 61b. The road to Gologuz Peak is 17 + 20.00 hm long. Its beginning is at 850 m above sea level, and the road continuously climbs to 1022.4 m above sea level, overcoming an altitude difference of 172.40 m. Based on the above data, it is possible to calculate the average slope of the horizontal route, which is 10.02%. However, ocular assessment determined (because the main/implementing project design of Lisac forest roads no longer exists) that certain parts of the route were also built with a higher slope.

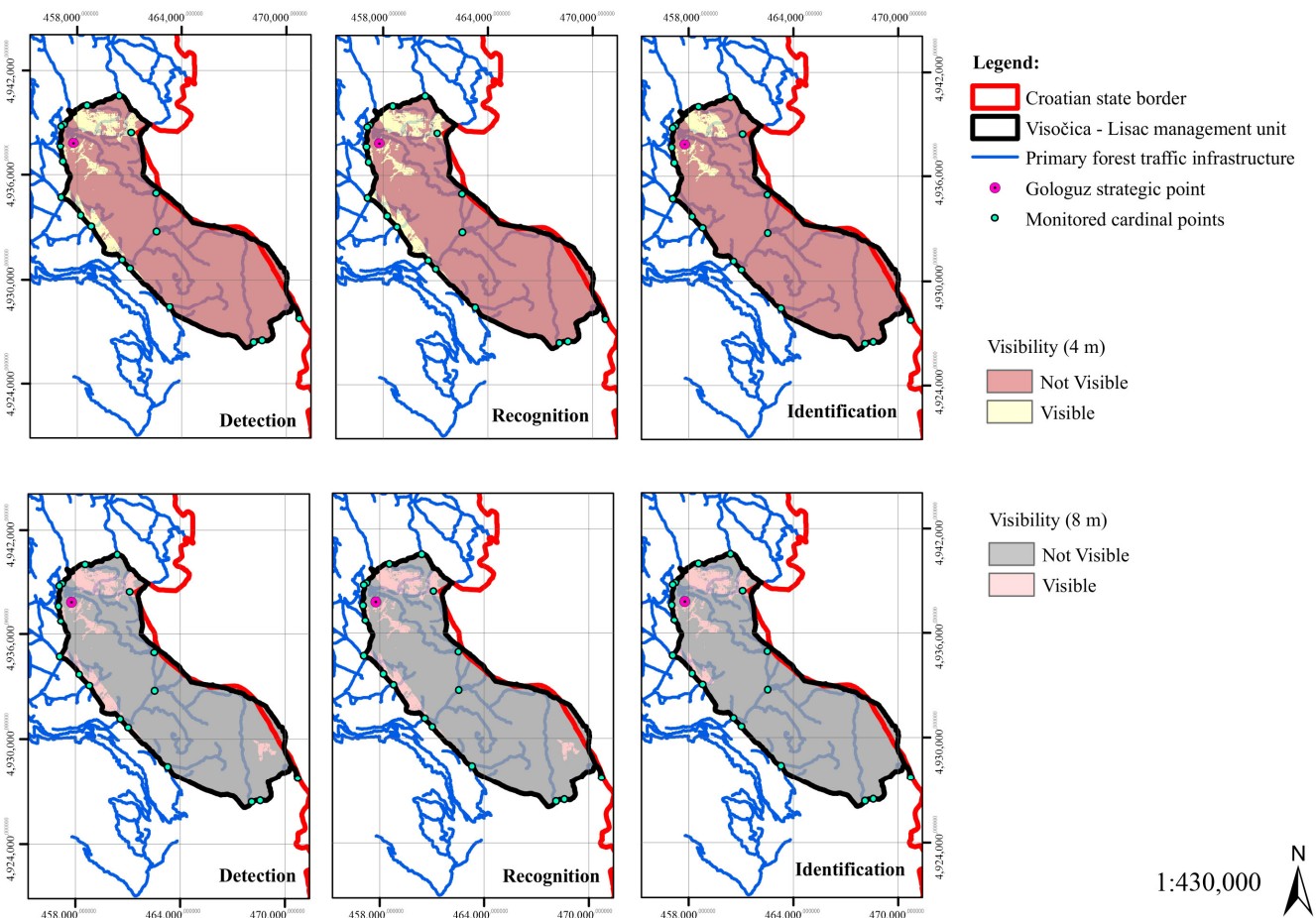

**Figure 3.** Visibility from the Gologuz strategic point by simulating the installation of a thermal imaging camera at a height of 4 m and 8 m. Source: Prepared by the authors.

When it comes to tactical surveillance points, taking into account their previously mentioned priority of forest protection, the research starts with the assumption that the position of such points depends on the position of cardinal points that are intended to be monitored in the road network. In this research, cardinal points are considered to be road intersections and the points of entry of each category of road into the area of the management unit. It should also be noted that they were all considered equally valuable.

Since tactical points are not significant for national security, it is considered that defining them is exclusively under the competence of foresters. Accordingly, the assumption is that a simpler, not sophisticated, video surveillance system will be installed at these locations. Therefore, a visibility analysis was conducted for the selected locations in such a manner that the stationary video surveillance system was set to half the maximum values from its technical specifications for the height of 8 m. The assumption is that a low-performance thermal imaging camera should be placed at a higher height in order to be able to additionally monitor a certain area.

The research results show that, from the proposed position for Tactical Point 1 (Pod Barićev Peak), it is possible to view (i.e., detect, recognize, and identify the vehicle) with four of the 14 marked cardinal points. Even if three cardinal points in the section dividing the management unit are excluded, it is concluded that the position of the tactical point is not satisfactory with regard to the surveillance of forest roads in the Visočica area. However, it is important to note that, from this position, it is possible to control most of the entry points to this part of the management unit (Figure 5).

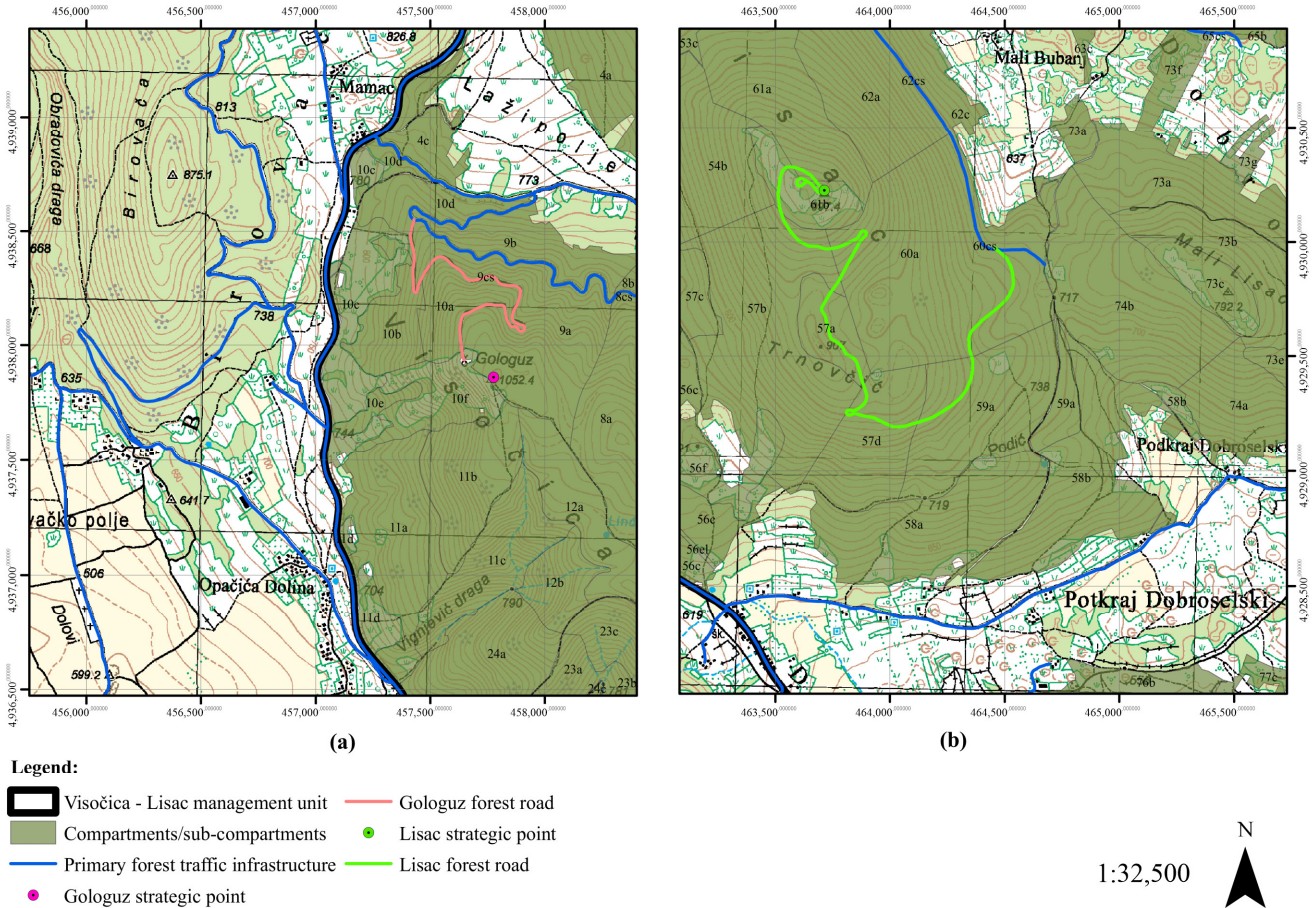

**Figure 4.** (**a**) Tactical forest road Gologuz and (**b**) Tactical forest road Lisac. Source: Prepared by the authors.

In the Lisac area, from the location of Tactical Point 2 (Špija Peak), of the marked eight cardinal points, it is possible to monitor four of them. It is important to mention that, from this location, it is possible to control most of the entry points to the Lisac area. Furthermore, from this point, it is possible to control the main forest road that divides the management unit in the north–south direction. In addition, visible is most of the section of the main forest road (in that area), which divides the management unit in the east–west direction (Figure 6).

This leads to the general conclusion that defining the position of the tactical point requires precise determination of the cardinal points to be monitored. In this case, it can be said that, from both tactical points, it is possible to monitor most of the entry/exit points to the management unit but not the key intersections within the road network. As the number of tactical points increases, the surveillance area increases proportionally, as [14] (p. 8) also wrote about.

When planning the conceptual routes of operational forest roads, three zero-line polygons were analyzed in the research. Given the position of Tactical Point 1, it has been established that it is convenient to approach this location from two directions (Figure 7a). Consequently, the length of the first conceptual route of the forest road (approaching the point from the south) is 722 m. From 790 m above sea level, it continuously climbs to Pod Barićev Peak (869.75 m above sea level), whereby it overcomes a height difference of 79.75 m. The average slope of the horizontal laying of the route is 10.09%, with a maximum of 12%. The calculated divider segment is 46.13 m. This route would mostly pass through the compartment/sub-compartment 5a and would slightly include the compartment/sub-compartment 4a. The second conceptual route, which approaches this tactical point from

the north, is 1095 m long. It overcomes an altitude difference of 74.75 m, i.e., it extends from 795 m above sea level to 869.75 m above sea level. The average slope of the horizontal laying of the route is 6.83%, with a maximum of 12%. The divider segment is 61.29 m. There are two points on this route where there is a change in slope. The first is located at an altitude of 825 m, and the second at 865 m. The forest road route thus planned would pass through four compartments/sub-compartments (3a, 4a, 4b, and 5a).

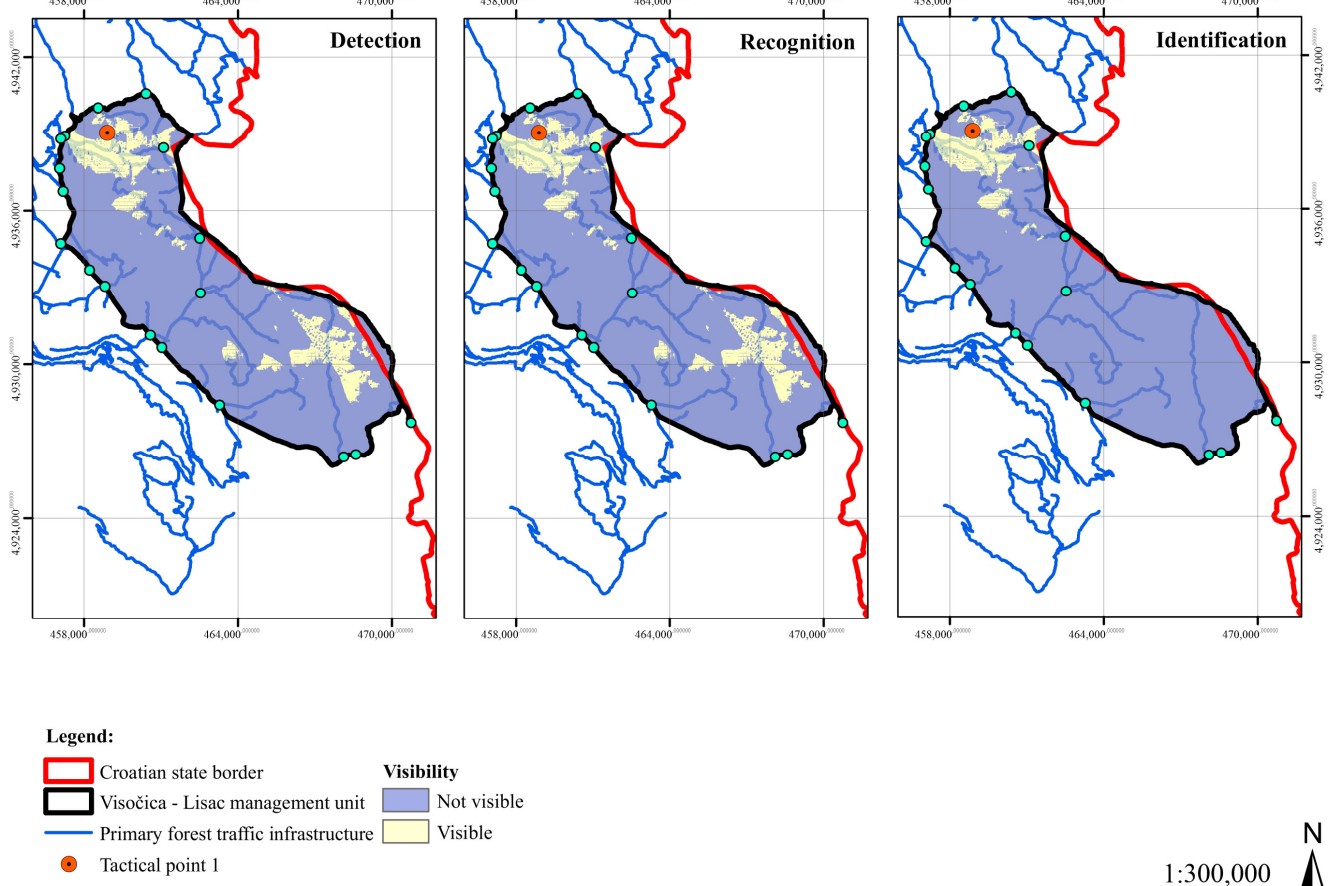

**Figure 5.** Visibility from Tactical Point 1 by simulating the installation of a thermal imaging camera at a height of 8 m. Source: Prepared by the authors.

Tactical Point 2 would be best approached from the north, along a 505-m-long route (Figure 7b). Its beginning would be at 770 m above sea level, and the end would be at Špija Peak at an altitude of 804.33 m. Consequently, this road would overcome an altitude difference of 34.33 m with an average slope of 6.80%, also not exceeding 12%. The calculated divider segment for this route is 26.51 m. The road would pass through three compartments/sub-compartments (37a, 37c, and 38a).

The results of the research on the planning of conceptual routes for operational multi-purpose forest roads have shown that they can be of a dual character. On the example of zero-line polygons for tactical surveillance points, it can be noted that such roads can be shorter or longer, depending on the position of the tactical point with regard to the existing road network and/or the quality of the accessibility of the part of the management unit where such a point is located. Therefore, it can be concluded that shorter routes should be planned in those parts of the management unit that are well accessible. Longer routes to the tactical surveillance point should be planned in inaccessible or poorly accessible parts of the forest complex.

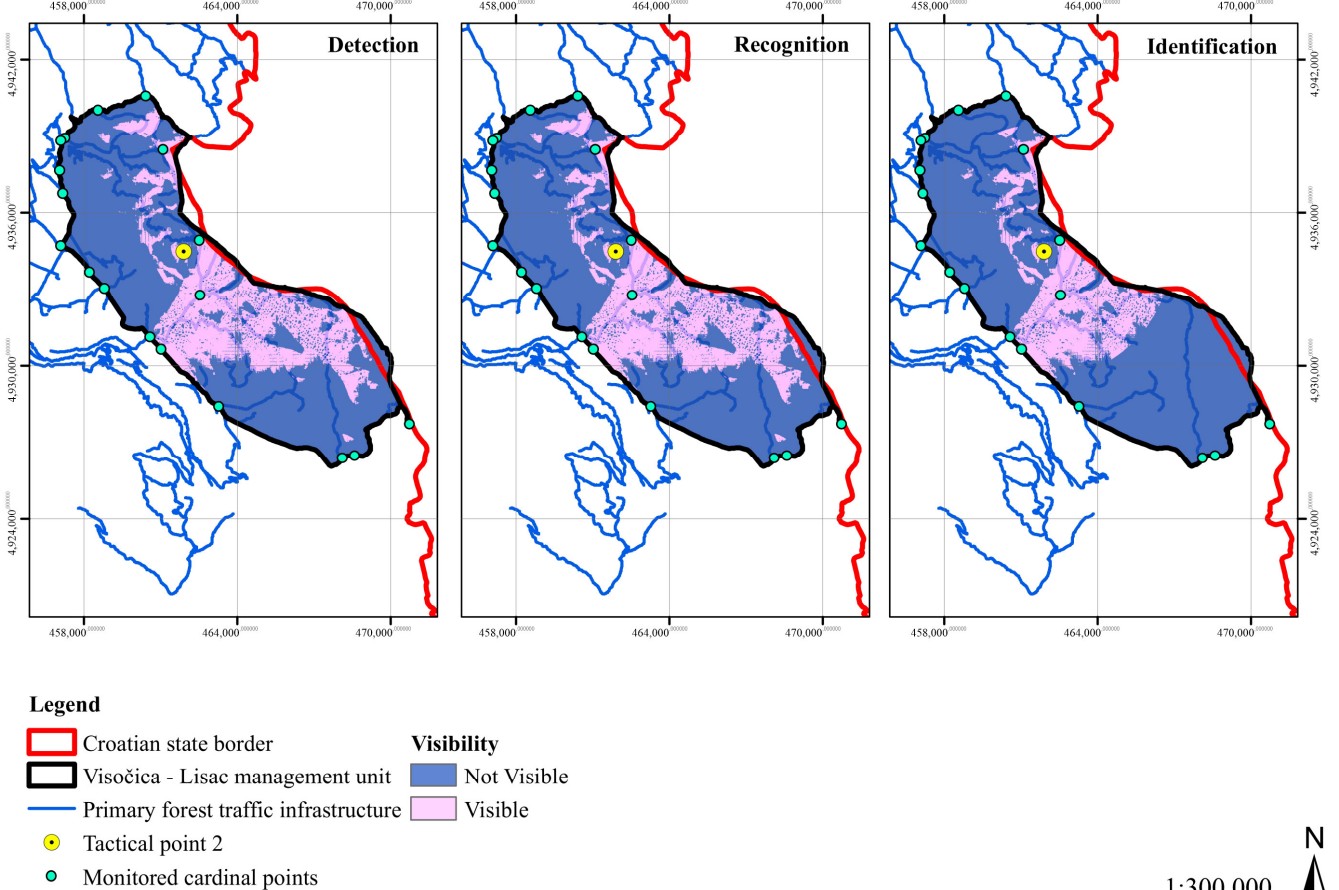

**Figure 6.** Visibility from Tactical Point 2 by simulating the installation of a thermal imaging camera at a height of 8 m. Source: Prepared by the authors.

In addition, during the analysis of the zero-line polygon in the Visočica area, it was observed that the total length of the route can consist of two parts. Therefore, for the purpose of monitoring the management unit, a shorter part of the route, which leads faster to the location of the tactical point, should be built first. Depending on its position, the assumption is that this part of the route will mostly pass through a smaller number of compartments/sub-compartments. For the purpose of forest management, a longer part of the total route of the operational road may be built first, assuming that it makes a larger number of compartments/sub-compartments more accessible. Until the construction of the full profile of the route, this part can be treated as a commercial forest road with a turning point at the end. However, further up to the elevation defined as the surveillance point, it is only possible to build a skid trail due to the possibility of using a larger longitudinal slope. In this manner, the surveillance point would be ready for activation at any time.

Based on all of the above, it is confirmed by the thesis of [40] (p. 24) that the optimal density of roads within a management unit cannot be discussed in general but that it is adapted to the micro-relief conditions of each management unit separately. However, it has been concluded that it is possible to plan two types of multi-purpose forest roads in the border management unit of a mountain area. The first type consists of short, tactical, or operational forest roads. The analysis determined that, in this management unit, such sections are between 500 and 800 m long, so it can be concluded that the forest roads in this example should be up to 1000 m long, depending on the appearance of the relief and the skill of the designer. In general, short multi-purpose forest roads have the characteristics of classic forest roads, as they separate from the main road towards the surveillance point and, by doing so, make a smaller part of the forest complex accessible (Figure 8a).

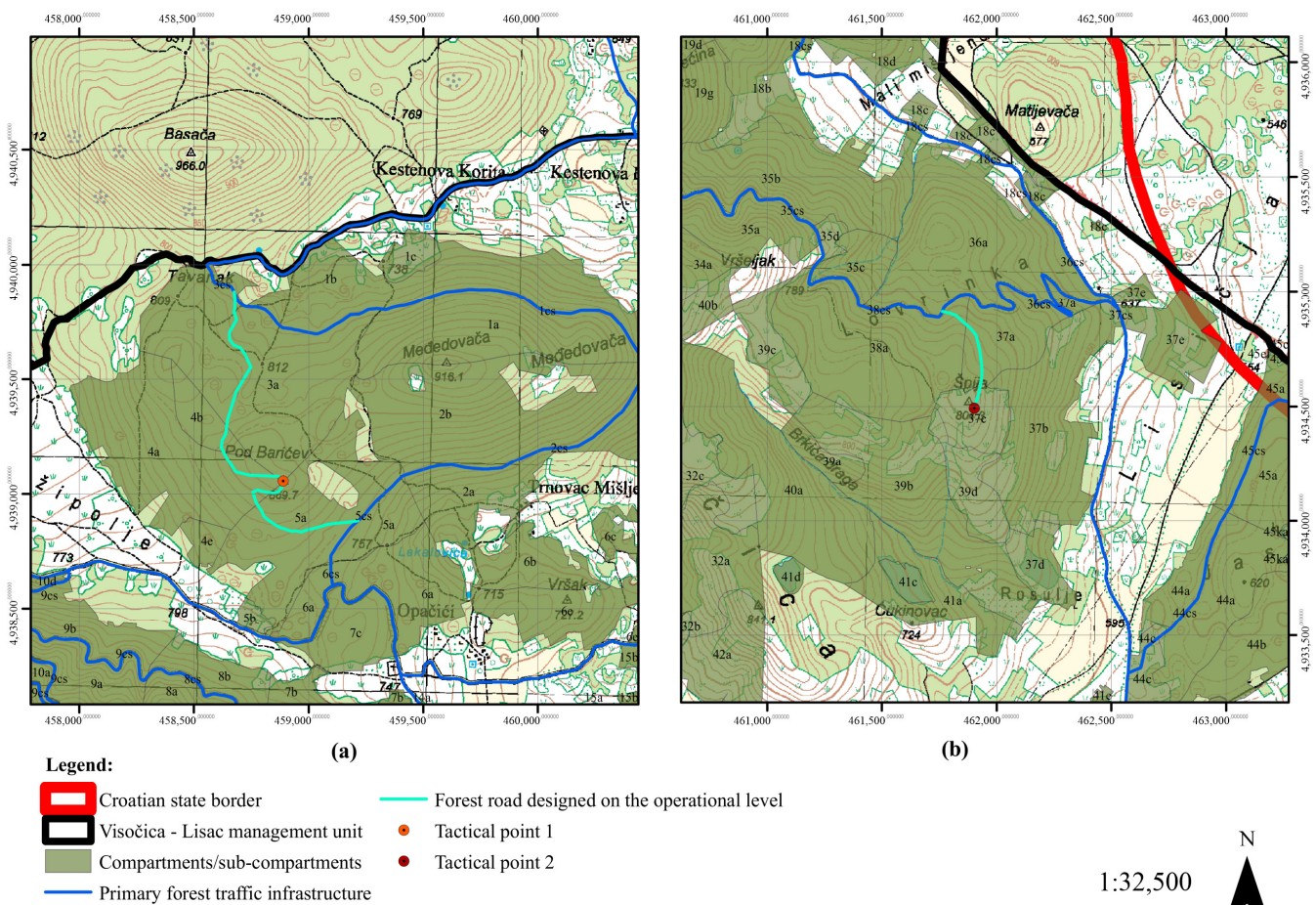

**(a)**　　　　　　　　　　　　　　　　　　　　　　　**(b)**

Legend:

▬ Croatian state border
▭ Visočica - Lisac management unit
▬ Compartments/sub-compartments
— Primary forest traffic infrastructure

— Forest road designed on the operational level
● Tactical point 1
● Tactical point 2

N

1:32,500

**Figure 7.** (**a**) Forest road designed on the operational level in Visočica area and (**b**) Forest road designed on the operational level in Lisac area. Source: Prepared by the authors.

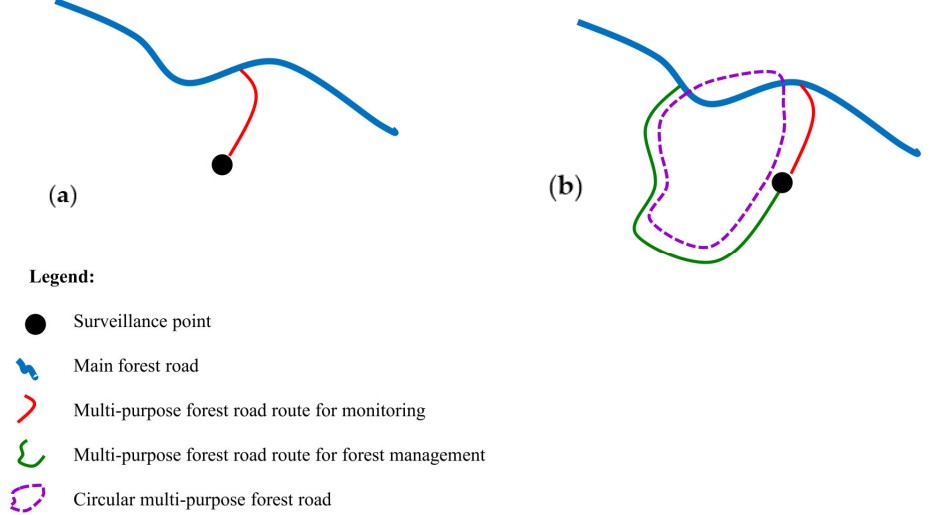

**(a)**　　　　　　　　　　　　　　　　　　**(b)**

**Legend:**

● Surveillance point

〜 Main forest road

⟩ Multi-purpose forest road route for monitoring

ʃ Multi-purpose forest road route for forest management

⬭ Circular multi-purpose forest road

**Figure 8.** (**a**) Classic multi-purpose forest road and (**b**) multi-purpose circular forest road. Source: Prepared by the authors.

The second type of multi-purpose forest roads are so-called circular roads (Figure 8b). The research found that operational forest roads belong more often to this group, but where possible, this is also feasible with tactical surveillance roads. For example, the assumption is that the Lisac (tactical) forest road is part of a route of such nature. The ar-

gument in favor of this is the fact that it currently passes through six compartments/sub-compartments. Assuming that a part of the circular route is built from the east, this road would make two additional compartments/sub-compartments accessible. The shorter route was probably not built first because the strategic point is positioned on a poorly accessible part of the management unit, so the benefits for forest management were also taken into account.

## 5. Discussion and Conclusions

Taking into account the integrity of the territory within the former Yugoslavia, some forest roads from Croatia continue to those in Bosnia and Herzegovina today. Therefore, the proposal for the establishment of a surveillance system in the mountainous border areas of Croatia is to control the road network in the management units closer to the state border. Based on the analysis of the existing, but also the possibilities for improving the future primary forest transport infrastructure in the Visočica–Lisac border management unit, the following has been concluded about the previously set research questions:

(1)  By simulating the surveillance of a management unit with two peaks at opposite ends, *the state border and the territory of the management unit can be controlled if the strategic surveillance point is positioned at the highest peaks of the management unit*. The analysis of the possibility of surveillance from Lisac Peak (the highest elevation in the south of the management unit) showed that, from this strategic point, it is possible to view the state border and half of the area of the management unit. From Gologuz Peak (which is not the highest elevation in the north), a satisfactory level of state border surveillance and of that part of the management unit is not achievable. However, from this point, it is possible to control deep into the Municipality of Donji Lapac.

(2)  The analysis has confirmed that *surveillance of forest roads within the management unit depends on defining the position of the monitored cardinal points and terrain relief.* The research found that a simpler video surveillance system would also be suitable for forest protection. By controlling the road network, not only criminal activities in the field of forestry would be prevented, but it would also contribute to strengthening national security in those border management units for which setting up a sophisticated video surveillance system is currently not envisaged. The only prerequisite for this is to draw attention to the position of the monitored cardinal points when defining the position of the tactical point in space. Namely, such a point is best placed on the highest elevation around which the largest number of cardinal points to be monitored are grouped.

(3)  It is confirmed that *the multi-purpose forest road designed on a tactical level affects the accessibility of the management unit, although this is not its primary purpose.* Analyzing the layout of the route of the Lisac (tactical) forest road, it can be noted that forest management was taken into account, although the road was built at the request of the Ministry of the Interior for state border surveillance. Namely, six compartments/sub-compartments, which had not been managed previously, were made accessible by its construction. Thus, the mean distance of skidding wood was reduced, as were the costs of forest exploitation. It is also worth mentioning that making accessible the compartments/sub-compartments with protective forests enabled better implementation of fire protection.

(4)  The analysis has confirmed that *the length of the multi-purpose forest road designed on the operational level affects the size of the area made accessible in a management unit.* The primary purpose of operating multi-purpose forest roads is to serve forest management; therefore, when planning them, one should always take into account the phytocoenological map of the management unit and inaccessible compartments/sub-compartments through which it would be favorable for the route to pass. The analysis found that, in the Visočica area, it would be desirable for the multi-purpose forest road to be longer because, in this manner, more compartments/sub-compartments

of the commercial beech forest would be made accessible, i.e., in phytocoenological terms, beech forest with autumn moor grass (Seslerio autumnalis-Fagetum sylvaticae (Horvat) M. Wraber ex Borhidi 1963) and mountain beech forest with large red dead nettle (Lamio orvalae-Fagetum (Horvat 1938) Borhidi 1963). However, for surveillance purposes, the route should be as short as possible, as the goal is to reach the tactical point in the fastest possible time. Consequently, in order to meet both requirements, circular roads should be planned, consisting of two routes: shorter (for surveillance) and longer (for forest management). On the other hand, the results of the analysis determined that the operational forest road in the Lisac area should be shorter, as it is defined in the well-accessible area of that part of the management unit. Consequently, its entire route can be used for both surveillance and forest management purposes.

The conducted research shows that the issue of green border control requires an interdisciplinary approach to the problem, which means that the forestry profession, in the process of establishing surveillance over forested areas along the state border, must not be ignored. Furthermore, in order to maintain a continuum in controlling the green state border, it has been concluded that the application of a two-level surveillance system is required in border management units. The higher level refers to the issue of national security. For this purpose, it is necessary to determine the position of one or more strategic surveillance points and to plan the construction of tactical multi-purpose forest roads that would lead to them. The lower level is intended to protect forests from activities primarily related to illegal tree felling, theft of wood assortments, poaching, and arson. For this purpose, it is proposed to determine the tactical surveillance points and plan the route of operational multi-purpose forest roads leading to them.

In further research on this topic, it is also worth considering the possibility of developing the categorization of primary forest roads for the border mountain area, given the priority of their surveillance for the purpose of forest protection but also for the control of the state's green border. By combining such a categorization of primary forest roads with the proposed two-level surveillance of forested areas, Croatia would have an elaborate system of not only controlling its green border in the mountainous area but could also control forest operations in border management units.

**Author Contributions:** Conceptualization, D.K.-T. and I.P.; methodology, D.K.-T. and I.P.; software, D.K.-T.; validation, I.P. and M.K.; formal analysis, D.K.-T. and I.P.; investigation, D.K.-T.; resources, M.K.; data curation, I.P.; writing—original draft preparation, D.K.-T.; writing—review and editing, I.P. and M.K.; visualization, D.K.-T.; supervision, I.P.; project administration, M.K.; funding acquisition, D.K.-T. and I.P. All authors have read and agreed to the published version of the manuscript.

**Funding:** This research received no external funding.

**Informed Consent Statement:** Not applicable.

**Data Availability Statement:** Data presented in this study are available upon request from the corresponding author. The data is not publicly available due to the protection of the state border.

**Conflicts of Interest:** The authors declare no conflict of interest.

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
