# Peer review of "Multi-Purpose Accessibility of Mountain Area Forests for the Purpose of Forest Management and Protection of the State Border"

_sustainability, doi:10.3390/su152416935_

Round 1
Reviewer 1 Report
Comments and Suggestions for Authors
In the system of planning road construction and monitoring border forests in mountainous areas, it makes sense to think about the large-scale use of unmanned aerial systems (UAS), integrated with automated pattern recognition software systems using neural networks, non-parametric modeling, artificial intelligence, etc., which have proven its high efficiency, in particular, during a special Russian military operation in Ukraine, to solve the problems of high-precision reconnaissance, guidance of artillery systems, detection and elimination of targets (in this case, illegal migration objects). Long-term (more than 10 years) practice of using UAS in forest management systems in different countries (detection of fires in forests, illegal logging, detection of outbreaks of forest pests, etc.) shows that these systems are very well integrated at the strategic and tactical levels of control forest management and state border protection, are effective in promptly obtaining reliable information along with ground control points, and are relatively inexpensive to use compared to traditional surveillance means, manned by aviation and ground means.
Author Response
The authors of the paper are aware of all the advantages and wide application of UAS in national security systems. Taking into account the comment of the peer-reviewer, the paper emphasizes that data on the application of such technology in the Croatian homeland security system is classified and not publicly available (p. 6). The authors are of the opinion that the Croatian security system is in a transition, that is, in a phase of combining conventional methods with modern technologies, which is additionally emphasized in the paper (p. 6)
Reviewer 2 Report
Comments and Suggestions for Authors
The paper presents a proposal for managing the roads in a forested area in the Balkans. The paper highlights the need for adequate use of roads to strengthen the area's surveillance.
The paper works on an interesting topic, but it needs to be improved to deliver the desired message. The paper is too wordy and needs to be better organized and focused. The reader comes to find the hypotheses of the paper in the methods sections. These hypotheses should be the core of the paper and should be well presented in the Introduction to guide the reader through the thought process of the authors. Reorganizing the paper should help the authors to deliver the message they wish. I have made some additional minor observations on the pdf file to help the authors obtain a more robust contribution.
As indicated earlier, the paper is wordy.
Author Response
Please find all our answers in the document attached.
